# A Systematic Review and Meta-Analysis of the Efficacy of Physical Activity Interventions among University Students

Fang Yuan [1], Sanying Peng [2,3,*], Ahmad Zamri Khairani [3] and Jinghong Liang [4]

1    College of International Languages and Cultures, Hohai University, Nanjing 210024, China; yuanf@hhu.edu.cn
2    Department of Physical Education, Hohai University, Nanjing 210024, China
3    School of Educational Studies, Universiti Sains Malaysia, Penang 11800, Malaysia; ahmadzamri@usm.my
4    Department of Maternal and Child Health, School of Public Health, Sun Yat-Sen University, Guangzhou 510080, China; liangjh78@mail2.sysu.edu.cn
*    Correspondence: pengsy@hhu.edu.cn

**Abstract:** Inadequate physical activity (PA) among university students, a growing concern, hinders their holistic health development and the formation of healthy habits. Current evidence regarding the impact of interventions to promote PA in this group remains inconclusive. Through a systematic review and meta-analysis, this study aims to evaluate the efficacy of PA interventions in promoting PA among university students. A search of six electronic databases up to 30 June 2023 was conducted to identify randomized controlled trials (RCTs) assessing PA interventions in university students. Thirty-one RCTs were included for systematic review and meta-analysis. The eligible studies' quality was assessed via the Cochrane Collaboration tool for evaluating the risk of bias. The results indicated a significant increase in total PA (TPA), moderate-to-vigorous PA (MVPA), and step counts at post-intervention compared to control groups, with effect sizes approaching large for MVPA and an increase of 19,245 steps per week. TPA also showed significant improvements during follow-up periods from three weeks to 12 months. Subgroup analyses revealed significant intervention effects across all subgroups, with the subgroups of post-COVID and sample sizes ≤100 showing larger effect sizes within their respective group. This review identified that interventions could enhance TPA, MVPA, and step counts among university students, with sustainable effects on TPA, while further research is needed for other outcomes. The university environment is conducive to PA interventions, and future interventions integrating e-health with in-person modes, coupled with matched theories and behavior change techniques, show promise. This review protocol has been registered on the platform of the International Prospective Register for Systematic Reviews (PROSPERO, CRD42023486749).

**Keywords:** physical activity; university students; health behavior; systematic review; meta-analysis

## 1. Introduction

The World Health Organization, synthesizing robust evidence from multiple studies, emphasizes the vital importance of regular physical activity (PA) in sustaining both physical and mental well-being [1]. Despite widespread recognition of its value, this health-enhancing behavior remains largely overlooked, especially among young people. A global survey revealed that over 81% of adolescents do not meet the WHO's PA recommendations [2]. Further research indicates a notable decline in regular high-intensity PA during early adulthood, particularly among university students [3]. A study involving more than 20,000 college students in the United States revealed that only 22.4% achieved the minimum PA guidelines [4]. In a diverse range of 23 countries, varying in economic status, 41.4% of university students exhibited insufficient PA, with percentages ranging from 21.9% in Kyrgyzstan to 80.6% in Pakistan [5]. This trend not only heightens risks of obesity, overweight, and diabetes among this demographic [6], but also reflects the educational system's limited role in encouraging PA [7]. As university students navigate the crucial transition

from adolescence to adulthood, cultivating healthy lifestyle habits is imperative for their long-term welfare [8,9]. Thus, the implementation of effective interventions to promote their PA is of paramount importance for health practitioners.

In this context, guided by overarching public health and educational policies, extensive trials have been conducted among university students to assess the efficacy of a wide array of intervention strategies [10–12]. These strategies include traditional approaches, such as diverse physical education courses [13–15] or structured PA assignments [16], and cognitive-behavioral modification techniques delivered via in-person lectures [17]. Moreover, the potential of digital or online interventions, leveraging social media [18,19], internet platforms [20,21], and electronic monitoring devices [22,23], has been explored, representing a shift towards information and communication technologies (ICT). However, these research endeavors have yielded mixed results. For instance, McDonough et al. [19] observed a significantly large effect size in moderate to vigorous physical activities (MVPA), in contrast to the findings of Schweitzer et al. [24], who reported no significant impact in similar exercise intensities. This disparity underscores the necessity for more comprehensive and varied evidential support to substantiate the effectiveness of PA enhancement measures in university students.

The current meta-analytic evidence does not converge on a uniform conclusion. Whatnall et al. identified significant impacts of interventions in enhancing step count, MVPA, and total PA (TPA) among young adults [10]. Contrastingly, Plotnikoff and colleagues, focusing on university students, noted significant effects only in moderate PA (MPA), albeit with a trivial effect size [25]. A meta-analysis by Favieri et al. [26], encompassing 18 randomized controlled trials (RCTs), reported a moderate but statistically insignificant effect size. These three quantitative meta-analyses represent the only comprehensive reviews to date that aggregate the impact of PA interventions on university students, offering valuable insights into this area. The review conducted by Whatnall [10], involving participants aged 17 to 35, provides relevant data for PA promotion in college students. However, the inclusion of individuals beyond the typical university age may introduce biases affecting the applicability of results to campus-based PA interventions. The trials included in Plotnikoff and colleague's meta-analysis [25] addressed outcomes targeting changes in other health behaviors, such as dietary behavior and weight loss, posing a challenge in accurately synthesizing PA outcomes within these diversified interventions. Favieri et al. [26] combined various measurements of PA into a single outcome, failing to quantify the effects based on the intensity and duration aspects of PA separately. This approach of conflating different PA outcomes could contribute to the observed high heterogeneity and might also impact the credibility of the evidence. Furthermore, these three studies focused solely on the immediate post-intervention effects, neglecting to investigate the long-term sustainability of these effects during follow-up periods. Additionally, a review by Masali et al. [11] offered a qualitative systematic evaluation of PA interventions for college students. While the study advocated incorporating high-quality, low-bias risk research to bolster evidence quality and highlighted the need for interventions to address a broad spectrum of determinants influencing PA, it notably lacked substantial quantitative analytical support. This indicates that conducting a quantitative meta-analysis of intervention effects aimed at enhancing PA among university students is essential for providing robust and reliable evidence to support PA promotion initiatives. Recent studies have brought new insights in the wake of heightened health awareness due to the COVID-19 pandemic. This evolving landscape necessitates employing more refined scientific methods and strict criteria for including high-quality RCTs. It is imperative to collate and assess the various PA outcomes immediately following the interventions and throughout follow-up periods. Such research promises more precise, comprehensive, and timely evidence to enhance PA among university students, advancing and augmenting prior meta-analyses.

Therefore, this systematic review and meta-analysis primarily aims to confirm the immediate and enduring impact of PA interventions on a diverse array of PA outcomes among university students. Another objective is to identify the main factors contributing to

heterogeneity through subgroup analysis and examine how intervention effects vary under different moderating conditions.

## 2. Material and Methods

### 2.1. Protocol and Registration Details

This review was performed following the updated version of PRISMA guidelines [27] and Cochrane Handbook for Systematic Reviews of Interventions [28]. This review protocol has been registered on the platform of the International Prospective Register for Systematic Reviews (PROSPERO, Registration number: CRD42023486749, https://www.crd.york.ac.uk/prospero/display_record.php?RecordID=486749 (accessed on 7 December 2023)).

### 2.2. Searching Strategies

In adherence to predefined search strategies (shown in the Supplementary Materials), relevant databases were queried, and the findings were compiled in Endnote 20 (Thomson ISI Research Soft, Philadelphia, PA, USA), a task completed in August 2023. The search encompassed six databases: PubMed, Embase, Cochrane, WOS, PsycInfo, and PsyArticle. The search strategies imposed no restrictions on the language and publication date.

The search parameters included "title, abstract, and keywords" or analogous terms, employing Boolean logic to effectively combine and organize the search terms. The search criteria were segmented into three key areas: (1) "college student", "university student", (2) "physical activity", "exercise", and (3) randomized controlled trials.

To ensure a thorough literature review and prevent overlooking pivotal studies, a recursive search was conducted on the references of related studies. This approach, known as the "snowballing" method, was utilized to systematically identify and follow up on additional relevant research—this comprehensive search methodology aimed to capture various studies pertinent to the research topic.

### 2.3. Eligibility Criteria

In accordance with the PICOS framework, which stands for participants, intervention, comparator, outcomes, and study design, the criteria for inclusion are outlined as follows:

Participants: University students registered and able to participate in PA were included. Students with disabilities and mental disorders were excluded. Obese or overweight students who can participate in physical activities independently were included. If the participants included university staff, the study was excluded.

Intervention: Intervention studies aimed at improving PA or exercise levels implemented in higher education environments are included. The intervention duration and follow-up periods were not restricted.

Comparator: Studies that did not implement intervention measures in the control group or only provided some educational guidance were included.

Outcomes: Any study that measured PA-related outcomes using subjective measurement questionnaires and objective measurement tools as the leading effect evaluation indicators were included, including step counts, various intensity physical activities such as TPA, MVPA, vigorous PA (VPA), MPA and Light PA (LPA), frequency, and time of participation in PA, energy expenditure, and other outcomes.

Study Design: To ensure the quality of the evidence from the combined included studies, only quantitative study designs of RCTs, which included pilot RCTs and cluster RCTs, were included in this study. Quasi-trials and non-RCTs were excluded.

### 2.4. Study Selection

After removing duplicate studies and those not meeting the review topic, two authors screened the remaining records based on their titles and abstracts. In the case of any discrepancies during the screening process, the third author would make the final decision.

### 2.5. Data Extraction

Relevant data from included studies were systematically extracted and stored in an Excel spreadsheet using a predefined data coding method. A pilot test with a subset of samples was conducted to ensure the efficiency and feasibility of the formal review. The extracted information included: study characteristics (i.e., sample size, proportion of female participants, country of experimentation); intervention details (i.e., content, theoretical basis, intervention, and follow-up durations); measurement tools and indicators for PA outcomes; intervention results for each group (i.e., pre- and post-intervention sample size, mean, and standard deviation).

In cases where a study involved multiple intervention groups, data from the group presumed to yield optimal results were exclusively extracted. Communication with corresponding authors was initiated to obtain necessary data for effect size amalgamation when not provided. Omitted coding information pertained only to studies where information extraction was incomplete. Two authors independently conducted data extraction, resolving discrepancies through consensus.

### 2.6. Quality Assessment

Guided by Cochrane's Risk of Bias Tool (version 2) [28], a thorough assessment covered seven methodological domains: (1) bias in the random selection process of experimental samples, (2) bias resulting from the concealment of the sample allocation process, (3) bias related to the concealment of participant allocation to groups, (4) bias resulting from the concealment of the outcome assessment process, (5) bias resulting from incomplete data outcomes, (6) bias resulting from selective reporting of outcomes favoring hypothesis interpretation, and (7) bias resulting from conflicts of interest.

Each study was rated three grades on assessing these seven domains: low-risk, unclear, and high-risk. In summarily assessing individual studies, given the inherent challenges in achieving complete blinding of outcomes in experimental studies related to PA, quality assessment is limited to evaluating the other six domains. Studies classified as high-risk exhibit either one domain assessed as high risk, or more than three domains assessed as unclear. Without high-risk domains and only two to three unclear items, a study is categorized as having a moderate risk. The study is designated as low-risk if there are no high-risk domains or only one domain with unclear risk. Bias risk assessments were performed using Revman 5.4.1 software (The Cochrane Collaboration, The Nordic Cochrane Centre, Copenhagen, Denmark). Two authors independently conducted the risk of bias assessment, with any discrepancies resolved through consultation with a third author.

### 2.7. Statistical Analysis

This review examined the quantitative pooled effect sizes of PA outcomes. The relevant data were extracted and analyzed using STATA 16.0 software (Stata Corp, College Station, TX, USA). When the mean and standard deviation data were not presented in the studies, they were converted by formulas using standard error, confidence interval, *p* value, and other results. Intention-to-treat analysis data were utilized when reported; otherwise, completer analysis data were employed. When there was no significant difference in PA measurements between the intervention and control groups at baseline, post-intervention measurements were included in the combination of quantitative effect sizes.

In evaluating the synthesized effect size for the intervention group, the steps indicator exhibited consistency in the unit of measurement across studies. The mean difference and 95% confidence interval were calculated using the random-effects model to pool the effect size. Due to variations in unit expression for indicators such as Total PA (TPA), Moderate-to-Vigorous PA (MVPA), Vigorous PA (VPA), and Light PA (LPA), the standardized mean difference (SMD) and 95% confidence interval, based on Cohen's d value and calculated using the random-effects model, were employed to determine the pooled effect size for the intervention group. Following Cohen's classification criteria [29], standardized mean

differences (SMD) below 0.20, 0.20–0.49, 0.50–0.79, and exceeding 0.80 correspond to trivial, small, moderate, and large effect sizes, respectively.

Qualitative visual funnel plots and quantitative Egger tests were employed to assess publication bias. Sensitivity analysis, utilizing the one-by-one elimination method, was conducted to evaluate the robustness of the pooled effect size. Cochran's Q Test and $I^2$ were used to test heterogeneity, with heterogeneity considered present when the Q test had $p < 0.05$ and $I^2$ values exceeding 50%, indicating moderate heterogeneity [30].

To explore influencing factors of heterogeneity and test moderating effects across various groups, subgroup analyses were performed on seven predefined categories: (1) Trial Period (Before COVID-19 vs. After COVID-19), (2) Region (Developed vs. Developing), (3) Intervention Mode (E-health vs. In Person), (4) Theory (Yes vs. No), (5) Duration (>5 W vs. ≤5 W), (6) Female Ratio (>50% vs. ≤50%), (7) Sample Size (>100 vs. ≤100).

## 3. Results

### 3.1. Study Selection

The search strategy yielded 11,407 records from five electronic databases potentially related to the research theme. After eliminating 1175 duplicates and discarding preliminary studies and records lacking sufficient research information, 121 articles were identified and included by referencing similar reviews. Consequently, 8997 studies underwent independent title and abstract screening by two authors. Subsequently, 132 publications and nine articles identified through manual search were selected for in-depth full-text review and further evaluation. Ultimately, 31 studies [14,17,19–24,31–53] were included in the systematic review and quantitative meta-analysis after excluding 69 studies that were either non-RCTs, lacked essential data, or failed to report critical results. Figure 1 comprehensively illustrates the complete literature selection process.

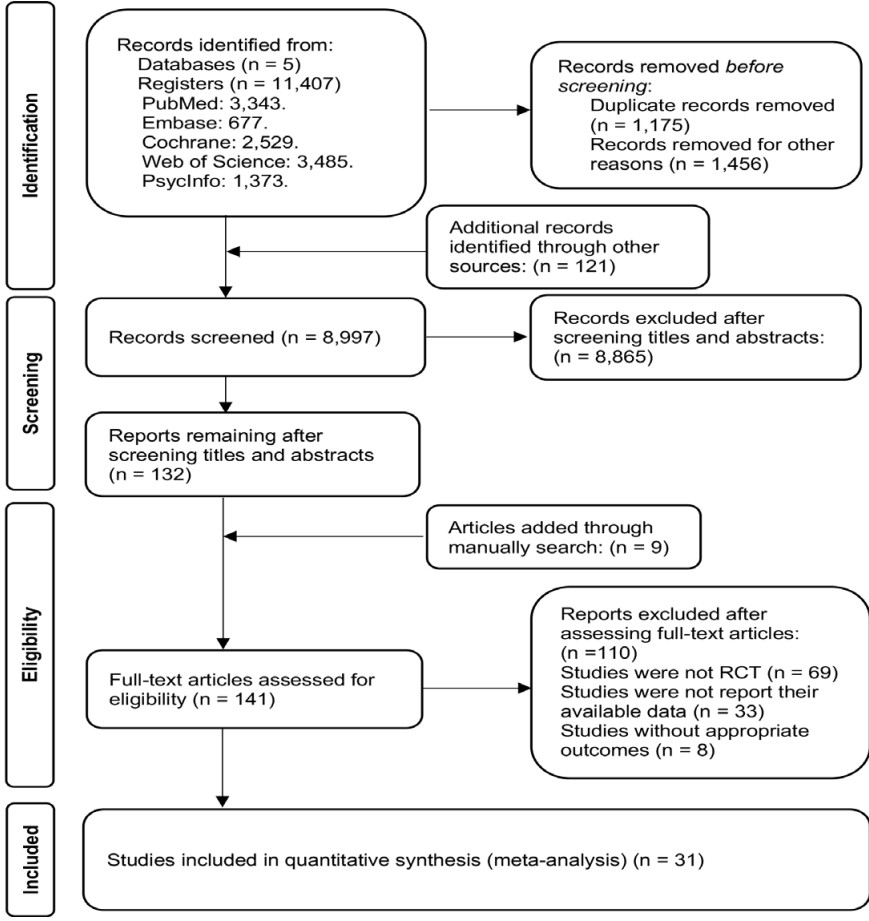

**Figure 1.** PRISMA flow chart of the literature selection process.

*3.2. Studies' Characteristics*

In the 31 included studies, 6872 university students participated, comprising 3481 in the intervention group and 3391 in the control group. The sample sizes of these studies varied, ranging from 12 to 1347, with a median of 123 participants. Three studies [44,45,53], exclusively involved female students, one focused on male students [51], and the remainder included participants of mixed genders.

Out of these studies, 25 were conducted in developed countries, with 18 based in North America—14 in the United States [14,19,22–24,31,32,37,39,41–43,48,49] and four in Canada [40,50,52,53]. Meanwhile, six studies [17,33,36,44,45,47] were undertaken in developing countries. Regarding the intervention models for PA, 21 studies [19–24,31,33–35,37–39,42,44,46,48–51,53] implemented e-health interventions, utilizing tools such as electronic health monitoring devices, internet platforms, and digital media. Meanwhile, ten studies [14,17,32,36,40,41,43,45,47,52] utilized traditional intervention methods such as exercise logs, training courses, and consultative interventions. Behavioral theories guided interventions in 13 studies [17,19,20,23,31,34,35,44,46–48,50,53], with the Social Cognitive Theory (SCT) being the most frequently employed.

Seven studies [17,20,36,39,42,46,53] measured PA outcomes at post-intervention and during follow-up periods, while other studies only assessed outcomes immediately at post-intervention. Intervention durations ranged from 4 weeks to 12 months, with follow-up assessments conducted between 5 months and 12 months.

Twenty-one studies [14,17,21,24,32–43,46,47,50,51,53] utilized subjective measurement questionnaires for assessing PA, with 13 of them employing the IPAQ [14,21,33–39,42,46,47,51]. Two studies utilized subjective and objective measurements [33,53], while others opted for objective measurement tools. PA outcomes measured across the included literature encompassed step counts, TPA, MVPA, MPA, VPA, and LPA.

The details of studies' characteristics shown in Table 1.

**Table 1.** Characteristics of the included studies.

| Study | Sample Size | | Country | Female (%) | Intervention Characteristics | | | Outcomes | |
|---|---|---|---|---|---|---|---|---|---|
| | IG | CG | | | Intervention Content | Theory | Duration/ Follow-Up | Instruments | Indicators |
| Al-Nawaiseh et al., 2022 [31] | 56 | 58 | USA | 80.70% | Theory-based smartphone app | Goal Setting | 12 W | Pedometer | Steps |
| Annesi et al., 2017 [32] | 52 | 32 | USA | 69.00% | IPACs | No | 5 W | GLTEQ | TPA |
| Barği, 2022 [33] | 15 | 16 | Turkey | 61.29% | PA counselling through distance learning | No | 4 W | Pedometer; IPAQ | Steps; TPA; VPA; MPA; LPA |
| Belogianni et al., 2023 [34] | 50 | 38 | UK | 67.05% | Online digital interventions using game elements | Gamification theory | 10 W | IPAQ | TPA |
| Brown et al., 2014 [52] | 28 | 32 | Canada | 68.33% | A residence community–based intervention | No | 20 W | GPAQ | MVPA |
| Cameron et al., 2015 [35] | 579 | 690 | UK | 55.36% | Online theory-based intervention | TPB; II | 1 M | IPAQ-SV | TPA |
| Choi et al., 2020 [36] | 188 | 184 | Hong Kong | 70.16% | Sport education within a compulsory physical education program | No | 10 W/15 W | IPAQ-SV | TPA |
| Diez et al., 2012 [17] | 31 | 42 | Mexico | 73.97% | Health-promoting intervention using cognitive-behavioral techniques | Health Promotion Model | 1 W/3 M | HPLP-II | TPA |
| Eisenberg et al., 2017 [37] | 40 | 41 | USA | 75.00% | Electronic behavioral monitoring (E-diaries and accelerometers) | No | 1 W | IPAQ-SV | TPA |
| Fukui et al., 2021 [38] | 39 | 49 | Japan | 54.40% | "Stay-at-Home Exercise" videos | No | 8 W | IPAQ-LV | TEE |

**Table 1.** *Cont.*

| Study | Sample Size | | Country | Female (%) | Intervention Characteristics | | | Outcomes | |
| | IG | CG | | | Intervention Content | Theory | Duration/ Follow-Up | Instruments | Indicators |
|---|---|---|---|---|---|---|---|---|---|
| Greene et al., 2012 [39] | 707 | 640 | USA | 35.00% | Online healthy eating and PA Program | No | 10 W/15 M | IPAQ-SV | TPA |
| Hall & Fong, 2003 [40] | 6 | 6 | Canada | 94.40% | A brief time perspective intervention | No | 3 W | A 30-day recall measure derived from the Stanford seven-day recall | VPA |
| Heeren et al., 2018 [41] | 91 | 85 | USA | 53.40% | Health-promotion intervention | No | 6 M | Three open-ended items | VPA |
| Kattelmann et al., 2014 [42] | 824 | 815 | USA | 67.20% | Twenty-one mini-educational lessons and e-mail messages (called nudges) | Yes/NR | 3 M/15 M | IPAQ-SV | TPA; VPA; MPA; LPA |
| Kim et al., 2018 [22] | 101 | 86 | USA | 62.03% | Wearable activity tracker in a credit-based PA instructional program | No | 15 W | ActiGraph Actitrainer | MVPA |
| Largo-Wight et al., 2008 [43] | 39 | 38 | USA | 62.00% | PA logs | No | 10 W | Health Canada and national quality institute questions | TPA |
| Lee et al., 2012 [44] | 46 | 48 | Taiwan | 100% | An intervention combining self-efficacy theory and pedometers | SET | 12 W | Pedometer | Steps |
| Lu et al., 2023 [45] | 59 | 58 | China | 100% | Tabata-style functional high-intensity interval training | No | 12 W | Accelerometer | TPA; MVPA |
| Maselli et al., 2019 [46] | 11 | 11 | Italy | 60.61% | Individual counselling and activity monitors | SCT + TTM | 12 W/3 M | IPAQ | TPA |
| McDonough et al., 2022 [19] | 32 | 32 | USA | 75% | A home-based, YouTube-delivered PA intervention grounded in self-determination theory | SDT | 12 W | Accelerometer | MVPA |
| Miragall et al., 2018 [20] | 26 | 26 | Spain | 85.50% | An internet-based motivational intervention | TTM | 3 W/3 M | Pedometer | Steps |
| Muftuler & Ince, 2015 [47] | 35 | 35 | Turkey | 42.86% | A PA course based on the trans-contextual Model | TCM | 12 W | IPAQ | TPA |
| Okazaki et al., 2014 [21] | 49 | 28 | Japan | 35.06% | An interactive internet-based PA intervention | No | 4 M | IPAQ | TPA |
| Peng et al., 2015 [48] | 25 | 23 | USA | 39.20% | An active video game | SDT | 4 W | Accelerometer | MVPA; LPA |
| Pope et al., 2019 [23] | 19 | 19 | USA | 73.68% | Wearable technology and social media | SCT + SDT | 12 W | Accelerometer | MVPA |
| Rote, 2017 [49] | 24 | 18 | USA | 47.62% | A Fitbit activity monitor | No | One semester | Pedometer | Steps |
| Schweitzer et al., 2016 [24] | 99 | 49 | USA | 68.24% | An electronic wellness program via email | No | 24 W | CCAPQ | MVPA |
| Sharp & Caperchione, 2016 [50] | 95 | 89 | Canada | 53.26% | A pedometer-based intervention | SCT | 12 W | The modified GLTEQ | VPA; VPA; LPA |
| Shin et al., 2017 [51] | 32 | 32 | Korea | 0 | SmartCare and financial incentives | No | 12 W | IPAQ, validated in Korean; | TPA |

**Table 1.** *Cont.*

| Study | Sample Size | | Country | Female (%) | Intervention Characteristics | | | Outcomes | |
|---|---|---|---|---|---|---|---|---|---|
| | IG | CG | | | Intervention Content | Theory | Duration/ Follow-Up | Instruments | Indicators |
| Sriramatr et al., 2014 [53] | 55 | 55 | Canada | 100% | A social cognitive theory-based internet intervention | SCT | 12 W/3 M | The Thai version of GLTEQ; Pedometer | Steps; TPA |
| Yan et al., 2023 [54] | 28 | 24 | USA | 80.77% | An eight-week peer health coaching intervention | No | 8 W | IPAQ | TPA; VPA; MPA; LPA |

Notes: CCAPQ: The Cross-Cultural Activity Patterns Questionnaire; CG: Controlled Group; GLTEQ: The Godin Leisure-Time Exercise Questionnaire; GPAQ: The Global Physical Activity Questionnaire; HPLP-II: Health-Promoting Lifestyle Profile YII; IG: Intervention Group; II: Implementation Intentions; IPACs: Instructional Physical Activity Courses; IPAQ: International Physical Activity Questionnaire; LPA: Light Physical Activity; M: Month; MPA: Moderate Physical Activity; MVPA: Moderate to Vigorous Physical Activity; PA: Physical Activity; SCT: Social Cognitive Theory; SDT: Self-Determinant Theory; SET: Self-Efficacy Theory; TCM: Trans-Contextual Model; TPA: Total Physical Activity; TPB: The Theory of Planned Behavior; TTM: The Trans-Theory Model; VPA: Vigorous Physical Activity. W: Week.

### *3.3. Quality of Included Studies*

Figure 2 illustrates the aggregated assessment of bias risk for each criterion. One study was identified as high risk in the randomization category, and another presented unclear descriptions of its randomization process. Regarding bias in reporting, all studies were deemed low-risk, a status achieved through preemptive measures in the inclusion criteria. Notably, in the context of outcome measurement blinding, all studies were classified as high-risk, acknowledging the near impossibility of concealing results in PA measurements, whether subjective or objective. Regarding selection bias, performance bias, and attrition bias, no studies were categorized as high-risk, although not all were classified as low-risk. Concerning other forms of bias, two studies were identified as high-risk.

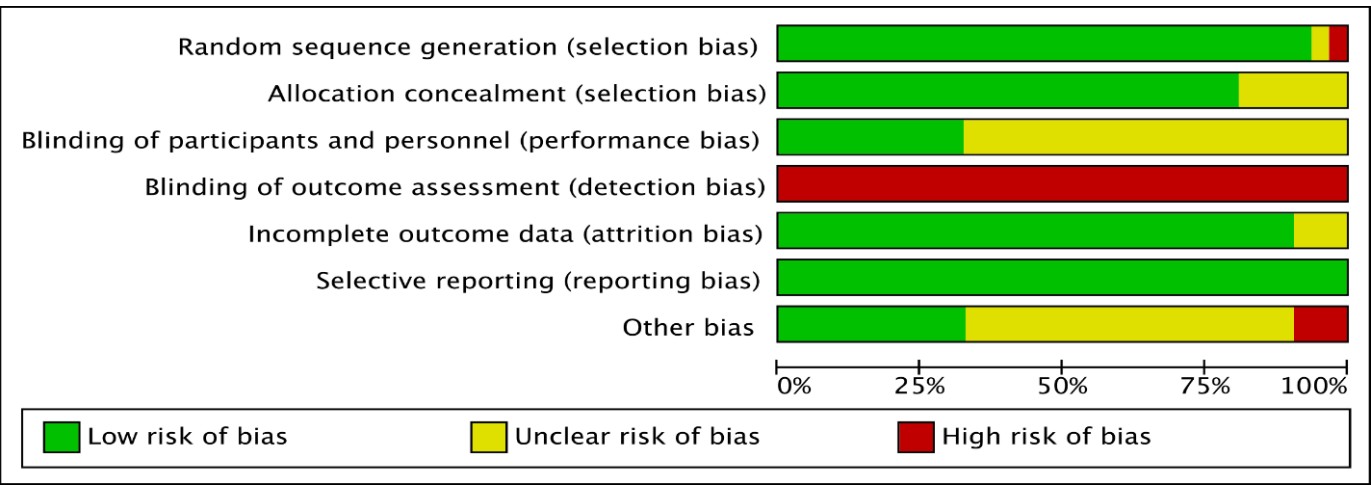

**Figure 2.** Risk of bias graph.

Figure 3 provides a detailed overview of each study's evaluation across various criteria. Based on these findings and in alignment with the risk assessment standards of this study, there were 14 studies classified as low-risk, 13 as medium-risk, and four as high-risk.

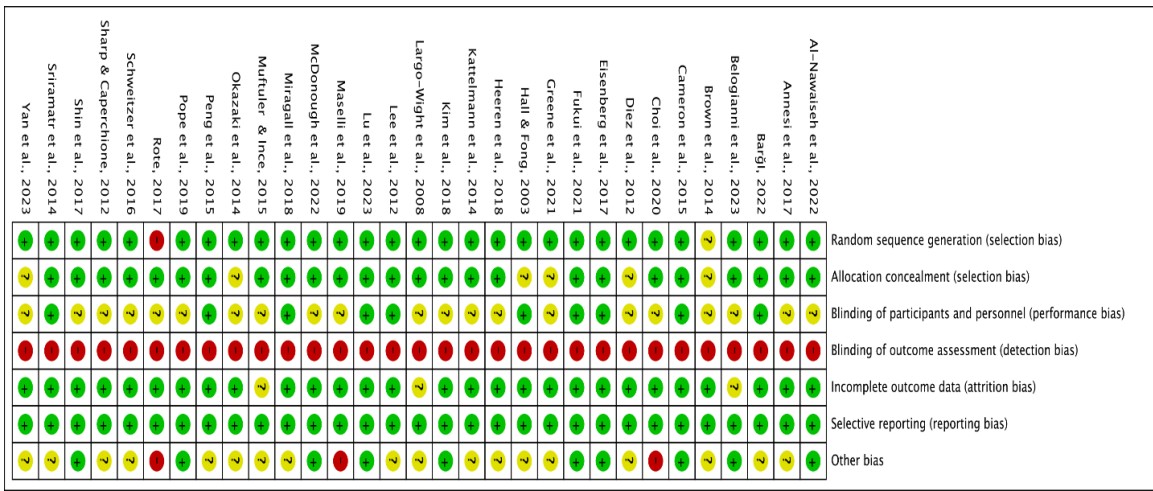

**Figure 3.** Risk of bias summary for include studies (green: low-risk; yellow: unclear; red: high-risk) [14,17,19–24,31–53].

### 3.4. Meta-Analysis

Utilizing a random effect model and the SMD, the synthesized effects on various PA outcomes at post-intervention are as follows: The PA interventions demonstrate a statistically significant small effect in TPA (SMD = 0.41, 95% CI: 0.27, 0.55, $p < 0.001$) (shown in Figure 4) and a significant effect in increasing MVPA (SMD = 0.74, 95% CI: 0.19, 1.29, $p < 0.001$) (shown in Figure 5), reaching an approximate high effect size level. However, the PA interventions do not yield significant effects on enhancing VPA (SMD = 0.14, 95% CI: $-0.03$, 0.29, $p = 0.08$), MPA (SMD = 0.18, 95% CI: $-0.01$, 0.36, $p = 0.10$) (shown in Figure 6), and LPA (SMD = 0.02, 95% CI: $-0.07$, 0.11, $p = 0.63$) (shown in Figure 7). Employing MD for the combined effect size of step count ((MD = 19,485.38, 95% CI: 10,008.34, 28,962.41, $p = 0.001$) (shown in Figure 8), the PA intervention group significantly increased weekly steps by 19,485.38 compared to the control group (shown in Figure 9).

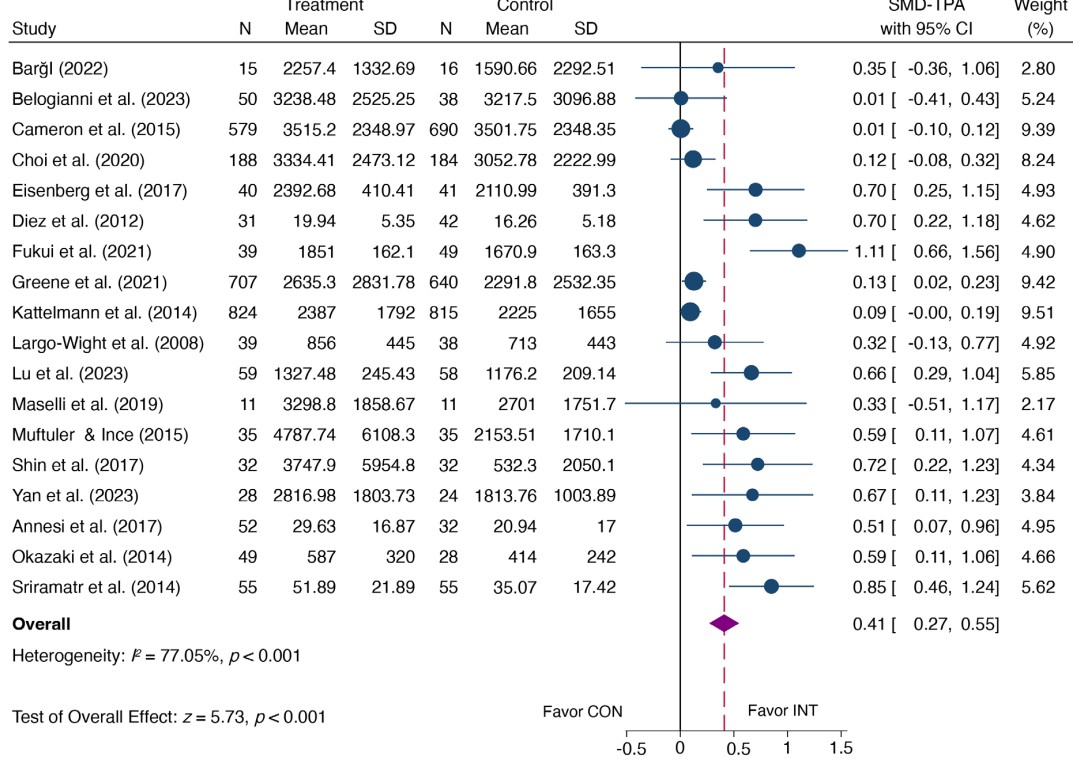

**Figure 4.** Meta-analysis of total physical activity at post-intervention [14,17,21,32–39,42,43,45–47,51,53].

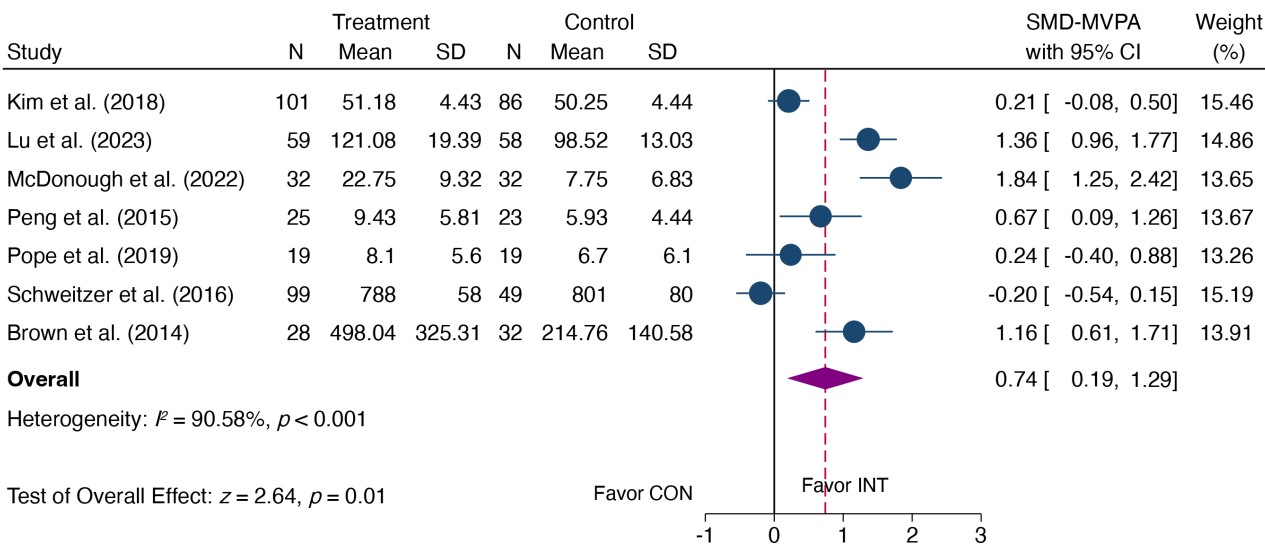

**Figure 5.** Meta-analysis of moderate to vigorous physical activity at post-intervention [19,22–24,45,48,52].

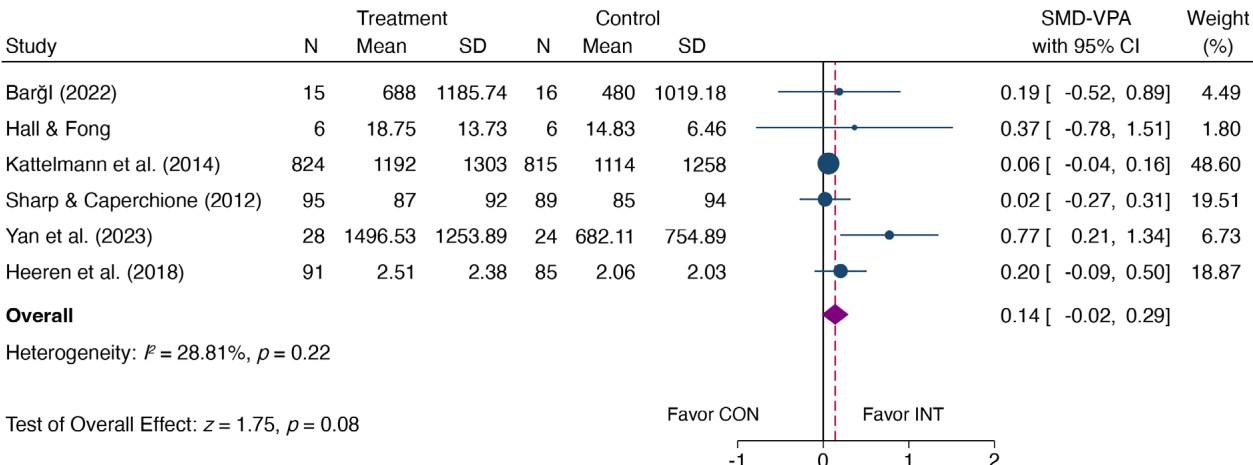

**Figure 6.** Meta-analysis of vigorous physical activity at post-intervention [14,33,40–42,50].

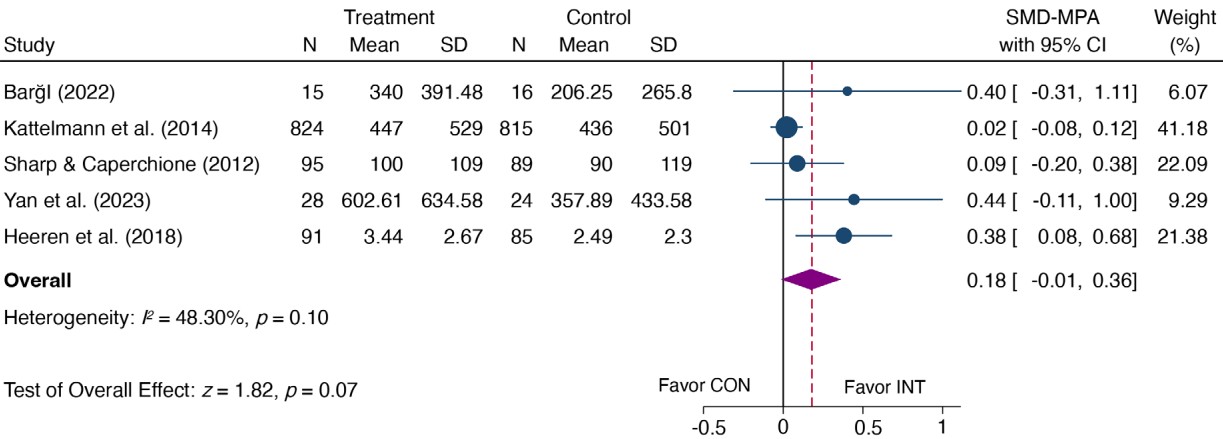

**Figure 7.** Meta-analysis of moderate physical activity at post-intervention [14,33,41,42,50].

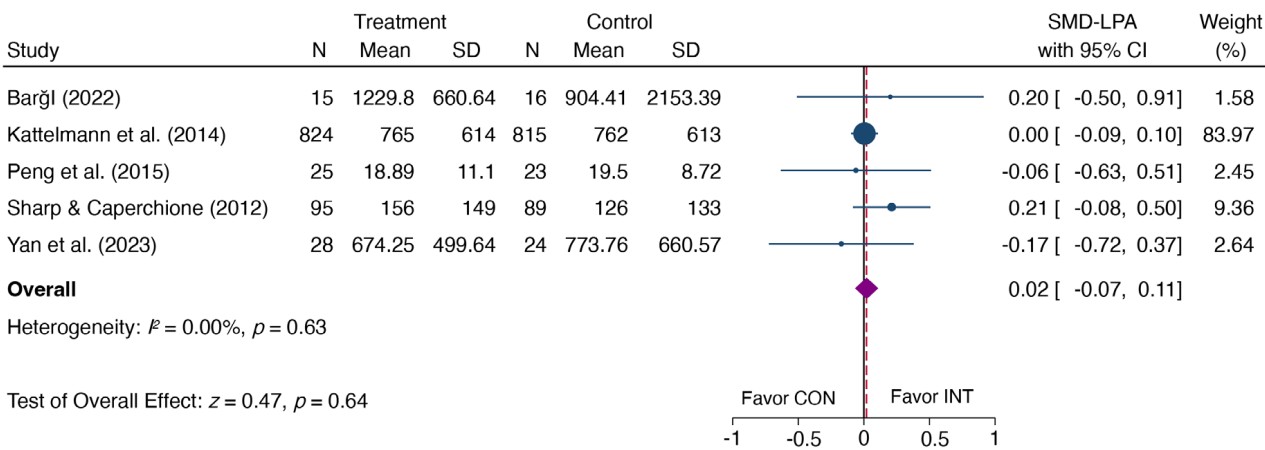

**Figure 8.** Meta-analysis of light physical activity at post-intervention [14,33,42,48,50].

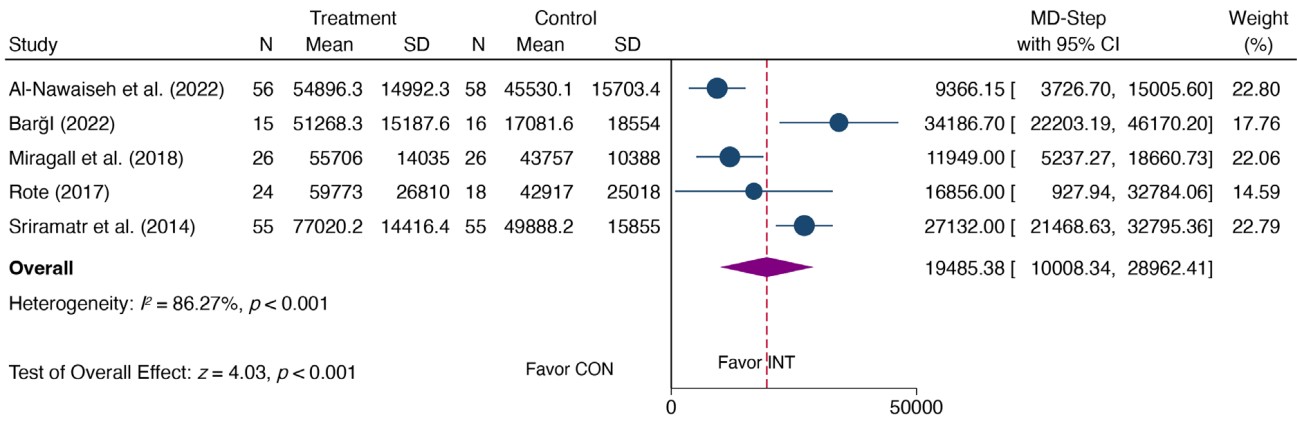

**Figure 9.** Meta-analysis of step count at post-intervention [20,31,33,49,53].

Applying a random effect model and the SMD, the synthesized effects of TPA during the follow-up period indicate a significant improvement in TPA (SMD = 0.49, 95% CI: 0.23, 0.75, *p* < 0.001) (shown in Figure 10), favoring the PA intervention group, approaching a medium-sized effect size. Due to fewer than three studies measuring other PA outcomes during the follow-up period, no meta-analysis was conducted for these outcomes.

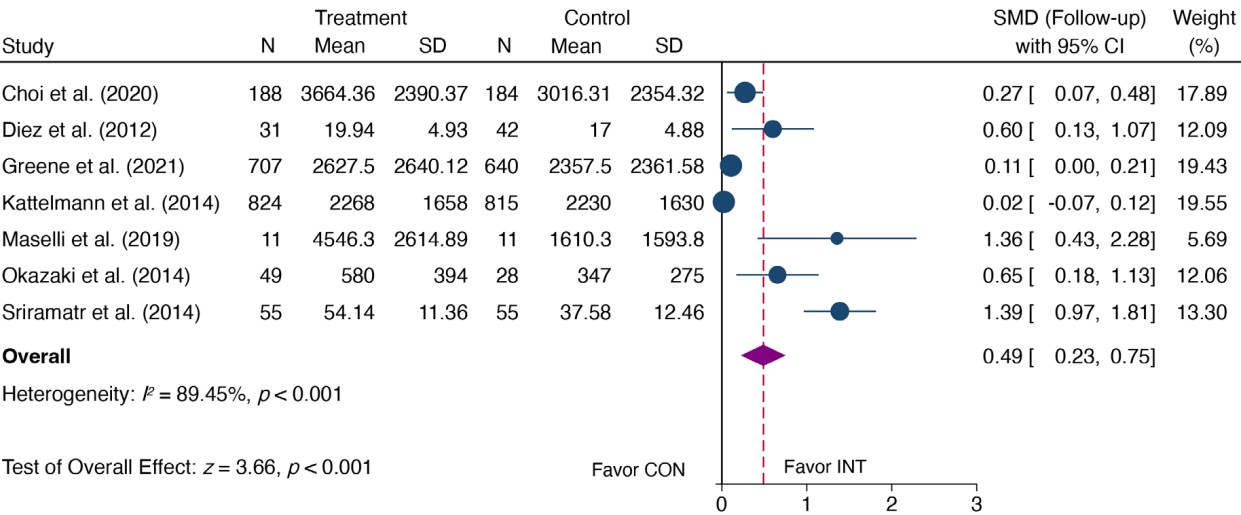

**Figure 10.** Meta-analysis of total physical activity at follow-up [17,21,36,39,42,46,53].

Furthermore, regarding heterogeneity in the synthesized results, no heterogeneity was observed for LPA at post-intervention, while moderate heterogeneity was found for MPA and VPA. High heterogeneity was identified for post-intervention TPA, MVPA, and step count. Similarly, high heterogeneity was found for TPA during the follow-up period.

### 3.5. Subgroups Analyses

Given the significant heterogeneity observed when synthesizing effect sizes for TPA, seven moderator variables that could be influential factors for heterogeneity were employed for subgroup analyses. The results in Table 2 illustrate the statistically significant effect of the PA interventions in all subgroups at post-intervention. Still, the post-grouping results do not indicate significantly less heterogeneity within subgroups in all subgroups. Heterogeneity disappeared in the developing country subgroup, as seen by the $I^2$ statistic of the subgroup analyses and dropped to a low level in the subgroups of post-COVID-19 and sample sizes $\leq$ 100.

**Table 2.** Subgroup analysis of total physical activity at post-intervention.

| Categories | Category | Studies | Heterogeneity Test | | SMD and 95% CI | $p$ |
| | | | $p$ | $I^2$ (%) | | |
|---|---|---|---|---|---|---|
| Trial Period | Post-COVID-19 | 14 | <0.001 | 72.00% | 0.31 (0.18, 0.45) | <0.001 |
| | After-COVID-19 | 4 | 0.276 | 22.50% | 0.75 (0.47, 1.03) | <0.001 |
| | Overall | 18 | <0.001 | 77.00% | 0.41 (0.27, 0.55) | <0.001 |
| | Between | | | 0.006 | | |
| Region | Developed | 14 | <0.001 | 77.90% | 0.36 (0.21, 0.51) | <0.001 |
| | Developing | 4 | 0.868 | 0.00% | 0.61 (0.38, 0.86) | <0.001 |
| | Overall | 18 | <0.001 | 77.00% | 0.41 (0.27, 0.55) | <0.001 |
| | Between | | | 0.066 | | |
| Intervention Mode | E-health | 11 | <0.001 | 80.90% | 0.37 (0.19, 0.54) | <0.001 |
| | In-person | 7 | 0.054 | 51.50% | 0.47 (0.25, 0.55) | <0.001 |
| | Overall | 18 | <0.001 | 77.00% | 0.41 (0.27, 0.55) | <0.001 |
| | Between | | | 0.488 | | |
| Theory | Yes | 7 | <0.001 | 78.20% | 0.30 (0.09, 0.50) | 0.006 |
| | No | 11 | <0.001 | 73.60% | 0.50 (0.29, 0.71) | <0.001 |
| | Overall | 18 | <0.001 | 77.00% | 0.41 (0.27, 0.55) | <0.001 |
| | Between | | | 0.172 | | |
| Duration | >5 W | 13 | <0.001 | 77.00% | 0.42 (0.25, 0.59) | <0.001 |
| | $\leq$5 W | 5 | 0.001 | 79.50% | 0.43 (0.05, 0.81) | <0.001 |
| | Overall | 18 | 0.0001 | 77.00% | 0.41 (0.27, 0.55) | <0.001 |
| | Between | | | 0.974 | | |
| Female Ratio | >50% | 13 | <0.001 | 80.70% | 0.45 (0.27, 0.63) | <0.001 |
| | $\leq$50% | 5 | 0.021 | 65.50% | 0.35 (0.08, 0.63) | 0.013 |
| | Overall | 18 | <0.001 | 77.00% | 0.41 (0.27, 0.55) | <0.001 |
| | Between | | | 0.572 | | |
| Sample Size | >100 | 6 | <0.001 | 80.70% | 0.26 (0.06, 0.36) | 0.006 |
| | $\leq$100 | 12 | 0.172 | 27.80% | 0.56 (0.39, 0.73) | <0.001 |
| | Overall | 18 | <0.001 | 77.00% | 0.41 (0.27, 0.55) | <0.001 |
| | Between | | | 0.002 | | |

There are significant differences in effect sizes between subgroups in both trial period and sample size groups, with interventions in the subgroups of post-COVID-19 and sample sizes $\leq$ 100 having significantly higher effect sizes than their counterparts.

### 3.6. Publication Bias and Sensitivity Analyses

Utilizing a funnel plot to examine the publication bias of TPA outcomes at post-intervention, the asymmetrical pattern depicted in Figure S1 suggests the presence of

publication bias. Subsequently, quantitative Egger's test results confirmed the existence of publication bias ($p < 0.001$). Egger's tests were further applied to other PA outcomes, including VPA ($p = 0.1587$), MVPA ($p = 0.3466$), MPA ($p = 0.0982$), LPA ($p = 0.8920$), and step count ($p = 0.4593$), as well as follow-up TPA ($p = 0.0154$). The Egger's test results indicated that publication bias was only evident in follow-up TPA.

A sensitivity analysis employing a stepwise exclusion of the literature was conducted, and the stability of the aggregated effect sizes was observed (shown in Figures S2–S8 of Supplementary Materials). Sensitivity analyses for the six post-intervention PA outcomes, including TPA, VPA, MVPA, MPA, LPA, step, and follow-up TPA, consistently demonstrated the summarized reliability and robustness of the summarized results.

## 4. Discussion

This systematic review and meta-analysis consolidates evidence from RCTs among university students, assessing the effectiveness of interventions to enhance PA. The categorized and aggregated results reveal that, compared to the control group, the intervention group exhibited a significant yet small effect size increase in TPA and a substantial, almost large effect size increase in MVPA, along with a significant rise in weekly step count by 19,485.28 steps. During the follow-up phase, the intervention group's TPA significantly improved, approximating a medium effect size. Furthermore, exploratory analyses of subgroups, defined by seven potential moderating variables, revealed significant intervention impacts across all subgroups. Notably, the post-COVID subgroup and the subgroup with sample sizes of 100 or fewer exhibited significantly greater effect sizes than their counterparts.

This review consolidates the significant impact of PA interventions, aligning with the findings of several similar meta-analyses conducted among the same age group population [10,54,55]. This provides robust empirical support for the effectiveness of PA interventions in university students. Importantly, prior reviews noted significant improvements in TPA, the most comprehensive measure of PA outcomes. This study's findings echo this, with intervention effects achieving a significantly small to moderate effect size, affirming that PA interventions can notably enhance university students' TPA. However, previous studies have also reported inconsistent results. For instance, a meta-analysis by Plotnikoff et al. [25] reviewed the effects of 18 TPA interventions targeting university students but did not report a significant increase in TPA. This discrepancy may stem from this study's focus on integrating PA with various health outcomes, encompassing literature from trials with diverse intervention goals. The significant findings reported here and in similar literature are restricted to studies where improving PA was the primary objective. This focus helps to mitigate the variability in intervention effects arising from multiple experimental goals and more accurately captures the effective evidence.

Regarding MVPA, it is widely employed by WHO and various health entities for monitoring PA and recommending guidelines. This study observed a significant intervention effect with an effect size nearing large. Similar findings were reported by Whatnall et al. [10] in young adults and Plotnikoff et al. [25] in university students, aligning with our results. Moreover, meta-analyses across different age groups, including adolescents [56], women [57,58], elderly people [59], and patients [60], have demonstrated significant improvements in MVPA due to interventions. This contrasts with another meta-analysis focusing on university students, where Favieri et al. [26] identified a medium effect size in MVPA enhancement through interventions. Still, the aggregate effect size lacked statistical significance. To resolve inconsistencies stemming from these aggregated results, synthesizing data using uniform measurement tools and establishing consistent eligibility criteria could reduce methodological heterogeneity, potentially mitigating discrepancies in the summarized findings.

Considering step count, only this study has aggregated intervention outcomes for this metric among university students, finding a significant increase of nearly 20,000 steps per week. In contrast, studies in general adults, such as those by Chaudhry et al. [61], Conn et al. [62], and Kang et al. [63], have reported a more modest weekly increase of

approximately 10,000 steps. Existing research proved that an increase of 1000 steps per day, or 10% of the recommended amount, is thought to be significantly associated with a reduced risk of all-cause mortality in adults [64,65]. WTO highlights that moving is good and that more PA leads to more incredible health benefits [1]. The findings of this study, then, that physical interventions can significantly increase step count, will be of great significance to the health promotion of university students.

The findings of this study on the intervention effects of VPA and LPA are consistent with the review by Plotnikoff et al. [25] in that no significant intervention effects were observed for either. Considering the limited scope of literature encompassing these two PA outcomes within this study, definitive and objective conclusions demand further investigation in future research. Importantly, there is a convergence in the measures of energy expenditure between LPA and step count. LPA is predominantly assessed through subjective questionnaires, whereas step count is quantified using wearable devices or smartphones equipped with pedometers. This methodological overlap may impair the precision in concurrently assessing these two outcomes in the same trial.

In addition, TPA also had a significant intervention effect over a follow-up period from 3 to 15 months, suggesting that the impact of the PA intervention can be maintained. This finding is also supported by previous studies [66–69]. PA maintenance is vital to developing behavioral habits but can quickly fade over time. Previous studies by Moeninginghoff et al. [68] and Murray et al. [69] also found that the maintenance of the effects of PA interventions is inversely proportional to time, which implies that PA behaviors should not be bout interventions but should be reinforced for a certain period after the intervention. Future research should delve into the optimal timing and methodologies for such reinforcement.

This study also aimed to identify the primary factors contributing to the notable heterogeneity observed in the combined effect size of TPA and to determine if group-based variables modulate these effect size disparities. The findings from seven subgroup analyses offer insightful evidence. Notably, the intervention effects within each of the seven subgroups achieved statistical significance, underscoring the effectiveness and potential applicability of PA interventions among university students. However, while heterogeneity within post-grouping subgroups was reduced to low levels or nullified in certain cases, this does not conclusively establish group-based factors as the predominant contributors to heterogeneity. Nevertheless, such exploratory analysis provides a foundation for future research.

The subgroup analysis revealed that, compared to their counterparts, subgroups formed post-COVID-19 and those with sample sizes of 100 or fewer exhibited significantly larger effect sizes. It is a well-acknowledged fact that COVID-19 drastically threatened global health. This pandemic fostered a widespread enhancement in health consciousness, particularly in recognizing and implementing the health benefits of PA [70], a trend distinctly pronounced among university students [71]. Thus, a pivotal factor for the augmented efficacy of post-COVID interventions could be attributed to the heightened awareness and consequent stronger behavioral motivation towards PA in this demographic. Furthermore, interventions targeting PA involve complex behavioral modifications. In smaller sample groups, there is tremendous potential for controlled implementation and adherence to the intervention protocols, thereby facilitating the achievement of anticipated intervention outcomes. These aspects may primarily account for the significantly greater effect sizes observed in the subgroups with sample sizes of 100 or fewer.

The subgroup analyses also found that the subgroups of developing countries, in-person intervention mode, no-theoretical-support, and more than 50% of females had larger effect sizes than their counterparts, but this difference was not statistically significant. Eliminating regional imbalances and advocating the active lifestyle of the general population is what WHO has been working on [72]. Trials have been conducted with better results in developing countries, which provides a rationale for disseminating PA intervention initiatives in these countries' universities. PA interventions are inexpensive,

but the health benefits are enormous and suitable for spreading in all developed and developing countries. ICT-enabled mHealth technologies have already achieved good effectiveness in PA promotion, validated by many relevant meta-analyses [12,58,68,73]. Then, this study has a weaker intervention effect in the remote mode than in the in-person mode. One possible reason is that face-to-face interventions on university campuses may be more effective in implementing intervention details. In contrast, university students who are already familiar with electronics and online media may be less motivated for this remotely delivered intervention. Following the meta-analysis by Yang et al. [54], which found that the combination of in-person and e-health yielded the highest PA intervention effects, future interventions should incorporate these two avenues of intervention.

Surprisingly, the no-theory group achieved larger effect sizes, which is inconsistent with the findings of Gourlan et al. [74]. Given the challenge of matching theories to specific measures during the intervention process, the theories utilized in many studies are dominated by cognitive and motivational enhancement theories such as SCT, SDT, etc., the nature of which is taught during the traditional educational process prior to PA interventions, and too much theory implantation may be counterproductive. Previous studies have also found that the type of theory employed [74] and the number of theories employed [10] did not have a definitive correlation with intervention effectiveness, with a single theory having even larger effect sizes. Therefore, PA interventions among university students should identify the determinants of behavior change and use matching theories to target the interventions. For example, if implementation intentions are proven to be effective, the action planning and coping planning embedded in them can be employed as efficient strategies for the specific interventions [75,76].

It is generally accepted that females are not as active as males in participating in PA and exercise based on an inherent sense of gender and static aesthetics. The present study's finding that the intervention was more effective in the group with a high percentage of females is consistent with Casado-Robles et al. [77], which conflicts with this notion but also implies that the motivation of females to participate in PA in university students may be related to their newer perceptions, such as the increasing awareness of the importance of maintaining a good image by exercising to control their weight and stay in shape [78].

There are many risks of non-communicated diseases and declines in physical fitness among university students that can be attributed to insufficient PA [79,80]. As university students move from second-stage education to tertiary education, they become more autonomous, but academic pressures, environmental discomfort and other unpredictable factors often make it increasingly complex for them to be physically active for the sake of their health, and some university students even have misconceptions about PA, which have been identified as barriers to promote PA [81]. Accordingly, promoting PA among university students by focusing on targeted interventions addressing key influencing factors based on existing evidence may yield promising benefits and represent a direction for future efforts.

This study exclusively included RCTs, ensuring a high quality of the selected literature. The rigorous eligibility criteria and selection process employed for literature inclusion, coupled with meticulous data extraction and analysis methods, provide a firm foundation for the credibility and robustness of the evidence presented in this study. However, there are inevitable limitations to consider. First, most of the RCTs incorporated in this study were conducted in North America, raising concerns about the broad representativeness of the sample. This limitation necessitates a cautious interpretation when generalizing and applying the study's conclusions. Second, despite synthesizing the homogenous PA of different intensities, the summarized outcomes still displayed notable heterogeneity. Furthermore, the limited number of studies included for some PA outcomes restricts further publication bias assessment and subgroup analysis. Third, the criteria for subgroup analysis were based on paradigms from previous studies and the researchers' interests. This arbitrary setting of moderating variables may have introduced bias into the results. Fourth, it is regrettable that the study did not differentiate and combine interventions of PA measured by self-reporting

and objective tools. Given that objective measurement tools, such as wearable devices and mobile apps, inherently contain elements of behavioral intervention through monitoring, exploring the differences in intervention efficacy between these tools and subjective measurement questionnaires is also of practical significance. Fifth, because some included trials utilized self-reporting and objective tools for measuring PA, a subgroup analysis was not conducted to differentiate these measurement modalities. Objective measurement tools, such as wearable devices and smartphone applications, inherently incorporate elements of behavioral intervention through their monitoring capabilities. Investigating the differences in intervention efficacy between these tools and subjective measurement questionnaires holds practical significance and could provide valuable insights for future research.

## 5. Conclusions

This systematic review and meta-analysis synthesized studies on the effectiveness of PA interventions for university students, identifying the significant impact of such interventions on TPA, MVPA, step count at post-intervention, and TPA at follow-up. These findings robustly demonstrate that PA interventions yield immediate and sustained effects, making them effective strategies for fostering healthy behavioral changes and long-term exercise habits in university students. Motivating students, enhancing self-efficacy, employing small sample interventions, and combining in-person with remote intervention modalities have proven to be effective strategies for optimizing intervention outcomes. University campuses, being hubs of educational and health experts and equipped with various facilities for living and exercising, provide the necessary conditions for cultivating positive lifestyles and healthy habits among students. The conclusions of this study provide solid theoretical support for the efficacy of PA interventions in university students, meriting consideration and application by policymakers and health practitioners.

**Supplementary Materials:** The following supporting information can be downloaded at: https://www.mdpi.com/article/10.3390/su16041369/s1, The search strategies, Figures S1–S8 and the PRISMA checklist are available online.

**Author Contributions:** Conceptualization, F.Y. and S.P.; methodology, S.P. and J.L.; formal analysis, S.P. and F.Y.; investigation, S.P. and F.Y.; writing—original draft preparation, F.Y. and S.P.; writing—review and editing, S.P. and F.Y.; supervision, A.Z.K.; project administration, S.P.; funding acquisition, F.Y. All authors have read and agreed to the published version of the manuscript.

**Funding:** This study was supported by the Fundamental Research Funds for the Central Universities (Hohai University, B230207048).

**Institutional Review Board Statement:** Not applicable.

**Informed Consent Statement:** Not applicable.

**Data Availability Statement:** Data generated or analysed in this study are included in the data tables listed in this published article or in the references. The original datasets can be obtained by consulting the corresponding author.

**Acknowledgments:** We thank all authors of the included literature for their contributions; We also thank the Department of Physical Education at Hohai University for facilitating this study.

**Conflicts of Interest:** The authors declare no conflict of interest.

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
