# Peer review of "A Systematic Review and Meta-Analysis of the Efficacy of Physical Activity Interventions among University Students"

_sustainability, doi:10.3390/su16041369_

Round 1

Reviewer 1 Report

Comments and Suggestions for Authors

The study is a systematic review and meta-analysis that evaluates the effectiveness of physical activity interventions for promotion among university students.

In the header of the paper, it should be specified that it is a review.

The abstract is structured and presents synthetically the important sections of the paper. It should be completed by specifying the quality assessment of the included studies. The methodology part should indicate the formed groups. The conclusion part should be more specific in relation to the content of the paper.

The introduction transposes the subject of the study and formulates the objective of the review at the end. However, before formulating the objective, it should be completed with research questions that the authors have identified by exploring the literature.

Regarding the methodology, there is room for improvement. The description should encompass the entire process, starting from study design and study area to study population, data analysis, and ethical clearance. The study period is not exactly indicated. In this sense, the material and methods section should present after the title and before the subsections a synthesis of the steps followed to orient the readers.

The actual subsections are well structured and provides all the information so that the study can be reproduced.

In line 199, the caption of table 1 should be reformulated. Notes from table 1 should indicate all abbreviations, for example IG, CG.

Figures 5, 6, 7, 8, 9 do not have a properly positioned caption.

In section 3.6 references are made to Figures S1, S2-S8 which cannot be identified in the text.

The discussion section analyzes its own results and compares them with other results reported in the literature. The section could be more concise to be easy to follow. At the end, the limitations of the study are presented. The section requires completion with some future directions of study.

The conclusions should better cover the entire content of the work.

The references are well written and recent.

Author Response

Dear reviewer,

Thank you for your comments concerning our manuscript entitled " A Systematic Review and Meta-Analysis of the Efficacy of Physical Activity Interventions among University Students" (sustainability-2853157). Those comments are all valuable and helpful for revising and improving our paper, as well as the essential guiding significance to our research. We have studied the comments carefully and have made corrections which we hope to meet with approval. Regarding the concerns raised by the reviewer, we have made every effort to provide thorough explanations to address them. Revised portions are marked in highlighted red (normal revision) in the paper using the track change mode of MS Word. The main corrections in the paper and the response to the reviewer's comments are as follows:

Point 1: “The abstract is structured and presents synthetically the important sections of the paper. It should be completed by specifying the quality assessment of the included studies.”

 Response1:

We appreciate your recommendation and have incorporated a description of the risk assessment into the abstract. For details, refer to line 15-16 on page 1.

Point 2: “The introduction transposes the subject of the study and formulates the objective of the review at the end. However, before formulating the objective, it should be completed with research questions that the authors have identified by exploring the literature.”

 Response2 :

We appreciate your recommendation and have add a highlighted description of research question into the introduction. For details, refer to line 71-73 on page 2.

 Point 3: “In line 199, the caption of table 1 should be reformulated. Notes from table 1 should indicate all abbreviations, for example IG, CG.

Figures 5, 6, 7, 8, 9 do not have a properly positioned caption.

In section 3.6 references are made to Figures S1, S2-S8 which cannot be identified in the text..”

 Response3 :

We appreciate your recommendation and have corrected all the issues of Table title and Figure. For details, refer to page 7-9, 11-13 and line 316 of page 14.

We are grateful for the reviewer's time and effort in evaluating our work, and we are committed to addressing their concerns and improving the manuscript accordingly. We welcome any further suggestions or guidance that will help us enhance the quality and impact of our research.

Thank you for your consideration.

Best wishes for you

Sincerely, yours

Reviewer 2 Report

Comments and Suggestions for Authors

Basic reporting

Dear authors, I’ve read with interest your work, and the topic is very interesting. The work, in my opinion, was conducted rigorously, and I have a few minor notes to make.

Abstract

I suggest to add this information:

-          Information was extracted using the populations, exposure, intervention, comparison, outcomes (PICOS) format

-          Bias evaluation tools utilized      

-          PROSPERO registration code

Keywords are usually different from those used in the main title

Introduction

The literature on the subject is sufficiently well summarised.

The abstract section represents a stand-alone section. Abbreviations given in the abstract should be restated, in extenso, in the rest of the manuscript, the first time they are used (e.g., physical activity at 28).

Once you use an abbreviation you must use it throughout the text, without alternating it with the unabbreviated version (e.g. physical activity at line 34).

Please check the whole manuscript.

Methods

The methods section is well described.

Validity of the findings

The results and discussion section are quite clear and organised. The parameters considered are well presented.

Minors:

You should check the citation style you’ve utilized. References should be numbered in order of appearance and indicated by a numeral or numerals in square brackets—e.g., [1] or [2,3]. Check the instructions for authors.

Comments on the Quality of English Language

Minor editing of English language required

Author Response

Dear reviewer,

Thank you for your comments concerning our manuscript entitled " A Systematic Review and Meta-Analysis of the Efficacy of Physical Activity Interventions among University Students" (sustainability-2853157). Those comments are all valuable and helpful for revising and improving our paper, as well as the essential guiding significance to our research. We have studied the comments carefully and have made corrections which we hope to meet with approval. Regarding the concerns raised by the reviewer, we have made every effort to provide thorough explanations to address them. Revised portions are marked in highlighted red (normal revision) in the paper using the track change mode of MS Word. The main corrections in the paper and the response to the reviewer's comments are as follows:

Point 1: Abstract 

I suggest to add this information:

-          Information was extracted using the populations, exposure, intervention, comparison, outcomes (PICOS) format

-          Bias evaluation tools utilized       

-          PROSPERO registration code.”

Response1 :

We appreciate your recommendation and have corrected all the issues mentioned above. For details, refer to line 15-16, line 24-25 of page 1, and line 108-109 of page 3.

 Point 2:Introduction 

The literature on the subject is sufficiently well summarised.

The abstract section represents a stand-alone section. Abbreviations given in the abstract should be restated, in extenso, in the rest of the manuscript, the first time they are used (e.g., physical activity at 28).

Once you use an abbreviation you must use it throughout the text, without alternating it with the unabbreviated version (e.g. physical activity at line 34). 

Please check the whole manuscript.”

 Response2:

We appreciate your recommendation and have conducted a thorough review of abbreviations across the document, ensuring they are abbreviated in accordance with established conventions.

Point 3: Minors:

You should check the citation style you’ve utilized. References should be numbered in order of appearance and indicated by a numeral or numerals in square brackets—e.g., [1] or [2,3]. Check the instructions for authors.”

 Response3 :

We are grateful for your recommendation and have accordingly revised the in-text citations and references in our manuscript to comply with the referencing style prescribed by MDPI publishers.

We are grateful for the reviewer's time and effort in evaluating our work, and we are committed to addressing their concerns and improving the manuscript accordingly. We welcome any further suggestions or guidance that will help us enhance the quality and impact of our research.

Thank you for your consideration.

Best wishes for you

Sincerely, yours

Reviewer 3 Report

Comments and Suggestions for Authors

A Systematic Review and Meta-Analysis of the Efficacy of Physical Activity Interventions among University Students

This research presents broad and ambitious aims. First of all, the main aim of the review is to verify the efficacy of Physical Activity interventions in promoting PA among university students. Subsequent objectives appear, complicating the investigation, given that the selection criterion “assessing PA interventions through randomized controlled trials (RCTs)” is not valid for subsequent variables.

In this quantitative research, the methodology is well explained in its steps and processes. References are updated. The article selection criterion: “only quantitative study designs of RCTs, which included pilot RCTs and cluster RCTs, were included in this study” is somewhat restrictive, creating a small problem. When you make categories such as sample size, countries, intervention characteristics, theories, duration, measuring instruments and indicators, a variety emerges that are impossible to align. It also results in the selected investigations taking place from 2008 to 2022, which is too long a period since the main changes in PA have occurred since 2016.

The data obtained (Table 1) focuses on study characteristics; intervention characteristics; measurement tools and indicators for PA outcomes. The table contains more than 18 acronyms! From this table, the authors wanted to present and obtain more analysis, which has resulted in a lot of figures and tables and, presumably due to lack of space, few comments. For example, in figure 2 and figure 3, the assessment of bias risk, we would have liked the explanation of the evaluation criteria used. Likewise, the meta-analysis of the PA interventions on outcomes consists of figures 4 to figure 10, which, as it does not have more than 3 lines of comments, represents an elucidative work on the figures for the reader. Next, we have the Subgroup analysis of total physical activity at post-intervention (table 2) with brief explanations despite containing numerous categories and analyses, ending with another brief comment on “Publication Bias and Sensitivity Analyses.”

In summary, it is patently a quantitative research in the classical canon, which sought to cover a number of variables that have not entered the selection criteria, which has resulted in variables that not have sufficient number of indicators and therefore, they do not support conclusions, as there is too much variation, as the authors themselves have mentioned when pointing out the limitations of the study. This happens, for example, when wanting to draw deductions about PA theories and perspectives. Likewise, some comments about PA in developed countries (25) and developing countries (6) are not justified since diversity does not allow comparison and the 6 that nominate "developing countries" are not on the UN list of developing countries.

Another additional observation is about the comment that “implies that the motivation of females to participate in PA in university students may be related to their newer perceptions, such as the increasing awareness of the importance of maintaining a good image by exercising to control their weight and “stay in shape.” This is not a difference, since there is research that indicates that men are motivated in a similar way to women to develop their muscles and get a gym body, the problem is that women have less time and more family obligations, as noted in the Lancet Glob. Health. 2018; cited by the authors.

It is a difficult theme, and many variables are not exactly quantitative. As the authors quote “The current meta-analytic evidence does not converge on a uniform conclusion”, including this research. However, we must recognize the authors' skill in quantitative methodology and encourage them to move towards more innovative perspectives.

Author Response

Dear reviewer,

Thank you for your comments concerning our manuscript entitled " A Systematic Review and Meta-Analysis of the Efficacy of Physical Activity Interventions among University Students" (sustainability-2853157). Those comments are all valuable and helpful for revising and improving our paper, as well as the essential guiding significance to our research. We have studied the comments carefully and have made corrections which we hope to meet with approval. Regarding the concerns raised by the reviewer, we have made every effort to provide thorough explanations to address them. Revised portions are marked in highlighted red (normal revision) in the paper using the track change mode of MS Word. The main corrections in the paper and the response to the reviewer's comments are as follows:

Point 1: “This research presents broad and ambitious aims. First of all, the main aim of the review is to verify the efficacy of Physical Activity interventions in promoting PA among university students. Subsequent objectives appear, complicating the investigation, given that the selection criterion “assessing PA interventions through randomized controlled trials (RCTs)” is not valid for subsequent variables.

 In this quantitative research, the methodology is well explained in its steps and processes. References are updated. The article selection criterion: “only quantitative study designs of RCTs, which included pilot RCTs and cluster RCTs, were included in this study” is somewhat restrictive, creating a small problem. When you make categories such as sample size, countries, intervention characteristics, theories, duration, measuring instruments and indicators, a variety emerges that are impossible to align. It also results in the selected investigations taking place from 2008 to 2022, which is too long a period since the main changes in PA have occurred since 2016.”

 Response1:

Thank you for your valuable suggestion. The inclusion of Randomized Controlled Trials (RCTs) in our quantitative intervention benefit trials is a strategic choice to ensure study quality, forming the foundation for the accuracy and reliability of the combined effect size evidence. The reviewer's reference to the primary changes in Physical Activity (PA) likely pertains to a comprehensive survey by Guthold et al. (2018) (in our manuscript’s reference list), illustrating the epidemiological scenario of PA prior to 2016, followed by extensive policy initiatives and interventional trials. Notably, PA interventions have been a continuous effort in health promotion and chronic disease prevention. Aggregating studies across interventions serves to avoid publication and selection biases, aligning with Cochrane's fundamental criteria for literature inclusion in meta-analyses. Consequently, we assert that including literature without temporal limitations is justified on theoretical grounds.

Point 2: “The data obtained (Table 1) focuses on study characteristics; intervention characteristics; measurement tools and indicators for PA outcomes. The table contains more than 18 acronyms! From this table, the authors wanted to present and obtain more analysis, which has resulted in a lot of figures and tables and, presumably due to lack of space, few comments. For example, in figure 2 and figure 3, the assessment of bias risk, we would have liked the explanation of the evaluation criteria used. Likewise, the meta-analysis of the PA interventions on outcomes consists of figures 4 to figure 10, which, as it does not have more than 3 lines of comments, represents an elucidative work on the figures for the reader. Next, we have the Subgroup analysis of total physical activity at post-intervention (table 2) with brief explanations despite containing numerous categories and analyses, ending with another brief comment on “Publication Bias and Sensitivity Analyses.”

Response2 :

We greatly appreciate the reviewer's attention to this matter. Given the numerous outcome indicators included in this study, we have endeavored to balance the comprehensive reporting of results with the need for brevity in the manuscript. Therefore, the study's findings are presented using succinct and precise language to effectively convey key insights within the constraints of the article's length.

Point 3: “In summary, it is patently a quantitative research in the classical canon, which sought to cover a number of variables that have not entered the selection criteria, which has resulted in variables that not have sufficient number of indicators and therefore, they do not support conclusions, as there is too much variation, as the authors themselves have mentioned when pointing out the limitations of the study. This happens, for example, when wanting to draw deductions about PA theories and perspectives. Likewise, some comments about PA in developed countries (25) and developing countries (6) are not justified since diversity does not allow comparison and the 6 that nominate "developing countries" are not on the UN list of developing countries.

Response3 :

We are grateful to the reviewer for highlighting the constraints of our study. Indeed, incorporating a diverse array of outcome indicators for Physical Activity (PA) presented a significant challenge in aggregating sufficient literature to generate more conclusive evidence. Despite these challenges, our study adhered to the established protocols of systematic reviews and meta-analyses, aiming to present the most relevant evidence available. This approach aligns with methodologies employed in prior related research, allowing us to not only update existing knowledge but also venture into new research domains. Regarding the classification of countries at the economic level, such as developing countries, we compared the United Nations classification systems of 2014 and 2021. The classification in this review conforms to these systems.

Point 4: “Another additional observation is about the comment that “implies that the motivation of females to participate in PA in university students may be related to their newer perceptions, such as the increasing awareness of the importance of maintaining a good image by exercising to control their weight and “stay in shape.” This is not a difference, since there is research that indicates that men are motivated in a similar way to women to develop their muscles and get a gym body, the problem is that women have less time and more family obligations, as noted in the Lancet Glob. Health. 2018; cited by the authors.”

Response 4 :

We are grateful for the reviewer's insightful suggestions regarding this aspect of our study. While traditional views hold that women, particularly in leisure activities, are less active than men, our subgroup analysis and the study conducted by Casado-Robles et al. observed a notably larger effect size in groups with a predominant female participation. Moreover, we referenced literature underscoring women's focus on self-image as a key motivator for engaging in physical activity, a trend more pronounced among younger women. This finding forms a critical part of our discussion and argumentation.

We acknowledge the prevailing situation: women, especially adults, often have limited leisure time due to family and childcare responsibilities, which restricts their opportunity for physical activity. The interpretations from our subgroup analysis are preliminary and exploratory, grounded in the evidence presented. However, we emphasize that further exploration of these relationships necessitates research designs that enable causal inference.

We are grateful for the reviewer's time and effort in evaluating our work, and we are committed to addressing their concerns and improving the manuscript accordingly. We welcome any further suggestions or guidance that will help us enhance the quality and impact of our research.

Thank you for your consideration.

Best wishes for you

Sincerely, yours